# An Experimental Study of a 3D Bone Position Estimation System Based on Fluoroscopic Images

**DOI:** 10.3390/diagnostics12092237

**Published:** 2022-09-16

**Authors:** Yuichi Yoshii, Yuta Iwahashi, Satoshi Sashida, Pragyan Shrestha, Hidehiko Shishido, Itaru Kitahara, Tomoo Ishii

**Affiliations:** 1Department of Orthopedic Surgery, Tokyo Medical University Ibaraki Medical Center, Ami, Ibaraki 300-0398, Japan; 2LEXI Co., Ltd., Sugamo, Tokyo 170-0002, Japan; 3Center for Computational Sciences, Tsukuba University, Tsukuba, Ibaraki 305-8577, Japan

**Keywords:** three-dimensional, tracking, computed tomography, fluoroscopy, preoperative plan, distal radius fracture

## Abstract

To compare a 3D preoperative planning image and fluoroscopic image, a 3D bone position estimation system that displays 3D images in response to changes in the position of fluoroscopic images was developed. The objective of the present study was to evaluate the accuracy of the estimated position of 3D bone images with reference to fluoroscopic images. Bone positions were estimated from reference points on a fluoroscopic image compared with those on a 3D image. The four reference markers positional relationships on the fluoroscopic image were compared with those on the 3D image to evaluate whether a 3D image may be drawn by tracking positional changes in the radius model. Intra-class correlations coefficients for reference marker distances between the fluoroscopic image and 3D image were 0.98–0.99. Average differences between measured values on the fluoroscopic image and 3D bone image for each marker corresponding to the direction of the bone model were 1.1 ± 0.7 mm, 2.4 ± 1.8 mm, 1.4 ± 0.8 mm, and 2.0 ± 1.6 mm in the anterior-posterior view, ulnar side lateral view, posterior-anterior view, and radial side lateral view, respectively. Marker positions were more accurate in the anterior-posterior and posterior-anterior views than in the radial and ulnar side lateral views. This system helps in real-time comparison of dynamic changes in preoperative 3D and intraoperative fluoroscopy images.

## 1. Introduction

Computerized virtual surgery planning has been increasingly applied in various orthopedic procedures. Three-dimensional (3D) preoperative planning and intraoperative navigation systems are clinically utilized for fracture management [1,2,3,4,5]. 3D preoperative planning is useful for pre-processing visualization, understanding fracture displacement, and selecting surgical procedures. 3D printed models and computer-assisted navigation also provide a secure way to successfully perform minimally invasive techniques. In our previous studies, we clinically applied the 3D preoperative planning system to the osteosynthesis of distal radius and humerus fractures [6,7]. It was found that 3D preoperative planning reduced postoperative correction loss and optimized implant selections [8,9]. The process of determining fracture reduction and placing the 3D models of the implants in virtual space aided decision makings during surgery. However, the reproducibility of the reduction position was only moderate [6]. Fracture reduction is technically challenging, and there were still gaps in implementing preoperative planning in actual surgery. This may be because there is no method for connecting the 3D preoperative planning image and fluoroscopic 2D image at the time of surgery.

Recently, the development of various applications for diagnosis using fluoroscopic image has been developed. In particular, the development of cross-modality applications, which actively use image information obtained by CT and MRI for diagnostic procedures using fluoroscopy, has made remarkable progress. There have been reports of applications that obtains a three-dimensional image of an object, and a method of superimposing on an X-ray fluoroscopic image to determine the additional information of anatomical positions of the organs [10,11,12,13]. Many of these applications have been reported in fluoroscopy using flat panel detectors. On the other hand, surgical fluoroscopy (C-arm), which is more often to be used in the orthopedic surgery, has not been sufficiently developed for com-bined modality applications with other imaging examinations due to image instability and distortion problems. This background led us to develop a system that connects the actual surgical field and 3D preoperative planning.

3D pose estimation is one of the most active topics in computer vision research. Effective algorithms exploiting 2D–3D point correspondences between pairs of images were introduced [14,15]. However, these techniques cannot be applied directly to transmission images (i.e., fluoroscopic images) as there often arises complications due to inconvenient calibration objects or failure of feature matching algorithms. A general goal of 2D–3D registration is to establish geometric transformation between the coordinate system of a 3D object and that of a device, such as a camera that captures a 2D image. In clinical use, it is important to align a 3D model of the anatomical structure that corresponds to the 2D radiographic image, which is typically obtained from regular X-ray, computed tomography (CT), or interventional fluoroscopy. 2D–3D registration methods were previously developed with several different protocols [16,17,18,19,20,21]. Some studies used edge-enhanced images of CT data, single- or bi-plane X-ray imaging with model-based shape matching, or projection images with tomosynthesis. These methods have been shown to be beneficial in situations where stable bone imaging is possible. However, techniques for aligning 3D images with the fluoroscopic images of structures that significantly move or deform during surgery have not yet been established. For example, in upper extremity surgery, it is necessary to confirm the reduction position and internal fixation installation position from various directions during surgery. It was considered that a new pose estimation system needs to be developed to compare 3D preoperative planning images with intraoperative 2D dynamic fluoroscopic images.

To compare 3D preoperative planning images and dynamic fluoroscopic images in real time, we developed a 3D bone position estimation system that displays 3D images created before surgery in response to changes in the position of fluoroscopic images during surgery. The objective of the present study was to assess the accuracy of the estimated position of 3D bone images with reference to fluoroscopic images. In our previous study, the differences between preoperative plan and postoperative reduction were about 2 mm, so we hypothesized that this system could depict 3D images compatible with fluoroscopic images with an error of less than 2 mm.

## 2. Materials and Methods

The study protocol was approved by our Institutional Review Board (T2019-0178). This is an experimental study of bone models. Radius forearm bone models of four conditions, normal, distal radius fracture, after fracture reduction, and after internal fixation, were evaluated with the bone position estimation system. Custom-made bone models were prepared based on CT data of past distal radius fracture cases. With using the CT data, the bone models were made from epoxy resin that could be visualized with fluoroscopy (Kyoto Kagaku Co., Ltd., Kyoto, Japan). A fracture model was created by osteotomy the bone model based on the fracture lines of the previous CT data. An orthopedic surgeon fixed the fracture with plates and wires to create reduction models. Bone models were covered with X-ray transparent elastic material (urethane resin) that imitated skin. The system estimates the 3D position of the forearm by comparing reference points on the fluoroscopic image with those on the 3D image created in the preoperative plan. The experimental setting was shown in Figure 1. The bone model was placed on the turntable to imitate the rotational movement of the forearm. To evaluate the accuracy of the 3D position of bone in the fluoroscopic image, a splint with four metal markers (markers 1–4) was placed on the radius bone model as reference points, and CT scans were performed. CT images were taken with a tube setting of 120 kV and 100 mAS, a section thickness of 0.8 mm, and a pixel size of 0.3 × 0.3 mm (Sensation Cardiac, Siemens, Germany). 3D bone images of the forearm models were created from the DICOM datasets of CT scans. Image analysis software (ZedView, LEXI Co., Ltd., Tokyo, Japan) was used to create a 3D bone image. After importing image data into the software, 3D images were created by extracting the bone lesion and reference points. A distal radius fracture was assumed, and a distal radius 3D model was created by extracting the area of the radius. The bone models were then visualized with fluoroscopy (Cios Select, Siemens, Germany). The C-arm of fluoroscopy was placed perpendicular to the bone model and the model was rotated to depict the bone image. The bone model was placed on the turntable and was placed the center of the X-ray output unit. The tracking of the position change of the bone model was verified by rotating the turntable. Bone positions were estimated from the reference points on the fluoroscopic image by comparisons to those on the 3D image.

### 2.1. 3D Position Estimation System

A program to detect reference points on the screen and track the motion of a fluoroscopic image was developed for the 3D position estimation and tracking. This 3D bone position estimation system is a program that outputs fluoroscopic images to a computer and can be operated on the computer. The algorithm of tracking is shown in Figure 2. The program was set up to track pre-specified reference points. In the present study, it was set to recognize metal sphere markers as the reference points. 1. CT images of bone models with splint were taken. At the beginning of the experiment, one fluoroscopic image was taken. The program automatically extracts candidate marker points from the image, and associates them with the points on the 3D image as the calibration. Automatic extraction of marker candidate points was performed by image processing. Circles with a certain radius were extracted by Hough transform. 2. Fluoroscopic images were monitored and candidate markers were automatically extracted from each frame. Automatic extractions of candidate markers were performed by the same image processing as in the first step. 3. Circular shapes detected by Hough transform were found more than the number of metal markers. Therefore, to determine the set of metal markers among them, a linear interpolation of the estimated positions from the last five frames was performed. 4. Refer to the result of linear interpolation and determine the nearest circular shape from it as the estimated position of the reference points in the next frame. 5. Compare the positional relationship of the markers on the fluoroscopic image in the current frame with the positional relationship of the reference markers on the 3D model. 6. Calculate the camera pose (orientation, position) that is closest to the positional relationship of the reference markers on the fluoroscopic image. As an index of proximity, the coordinates when a point on the 3D model is projected two-dimensionally with a certain camera pose were calculated, and then the pose that minimizes the squared errors of the distance of the reference markers were detected from the fluoroscopic image. 7. Finally, display the 3D image corresponding to the viewpoint of the 3D model.

The algorithm of the tracking was designed to track pre-specified reference points. The numbers represent the steps in the text.

### 2.2. Evaluations

To evaluate the accuracy of the estimated 3D position of bone models, the positions of the markers on the fluoroscopic image and on the created 3D bone image were compared (Figure 3). We verified whether the 3D bone image can be drawn by tracking positional changes in the forearm models. Accuracies were investigated by comparing the distance between markers on the fluoroscopic image (A) and on the 3D image, which was projected on the monitor (B). The distances between markers were measured using Image J software (NIH, USA). After importing the images into the software, measurements were performed under the following four conditions: posterior-anterior view, ulnar side lateral view, anterior-posterior view, and radial side lateral view. Differences in the distance of markers on the fluoroscopic image and on the 3D image (=A-B) were evaluated at each position. The distances were measured between marker 1-2, 2-3, 3-4, and 4-1. In addition, the intraclass correlation coefficients of marker distances between the fluoroscopic image and 3D bone image were assessed at each position. Averages of five times measurement for each model were used for further analysis. Differences in the measurements for each position were compared with a one-way analysis of variance. Multiple post hoc comparisons were performed using the Tukey honest significant difference test. All results were expressed as the mean ± standard deviation. Measurements were considered to be significant when the *p*-value was less than 0.05. All analyses were performed using SPSS Statistics (IBM, Tokyo, Japan) software.

## 3. Results

Correlations of marker distances between measurements on the fluoroscopic image and estimated 3D image are shown in Figure 4. Intra-class correlations coefficients of measurements between the fluoroscopic image and 3D image were 0.99, 0.98, 0.99, and 0.98 for the anterior-posterior view, ulnar side lateral view, posterior-anterior view, and radial side lateral view, respectively. All measurements showed excellent correlations between the fluoroscopic image and estimated 3D image.

Differences in marker distances between the fluoroscopic image and estimated 3D image are shown in Figure 5 and Figure 6. Average differences between measured values on the fluoroscopic image and 3D bone image for each marker corresponding to the direction of the bone model were 1.1 ± 0.7 mm, 2.4 ± 1.8 mm, 1.4 ± 0.8 mm, and 2.0 ± 1.6 mm in the anterior-posterior view, ulnar side lateral view, posterior-anterior view, and radial side lateral view, respectively (Figure 6a). The ratios of differences in the actual values (measurement difference/actual measurement in the fluoroscopic image) were 2.5 ± 1.8%, 9.0 ± 5.7%, 3.1 ± 2.2%, and 7.9 ± 7.0% in the anterior-posterior view, ulnar side lateral view, posterior-anterior view, and radial side lateral view, respectively (Figure 6b). Differences in the anterior-posterior and posterior-anterior views were significantly smaller than those in the ulnar and radial side lateral views (*p* < 0.05).

## 4. Discussion

In this study, a 3D bone position estimation system based on fluoroscopic images was reported. Tracking and navigation approaches generally include reference markers placed on patients or intervention devices. It allows preoperative image enrollment in intraoperative coordinate frames. Previous studies attempted 2D-3D registration for preoperative 3D images and fluoroscopy for intraoperative guidance [22,23,24,25,26,27]. These studies used either the initial calibration, the geometry approximated from the source-detector distance recorded in image data, or the geometry measured by the built-in measuring device. Another approach to geometric calibration is to image the patient together with a calibration fiducial of a known shape. However, these techniques are difficult to apply directly in orthopedic trauma surgery because of the additional costs, equipment, time, and lack of support for changes due to surgery. 2D fluoroscopic images are still the mainstream for orthopedic trauma surgery [28]. In the treatment of fractures, the affected area is largely deformed and moved by the surgical procedure, and a method to track this has not yet been established. It was necessary to consider the tracking of dynamic fluoroscopic images due to patient motion or surgical manipulation.

In the present study, we developed a 3D bone position estimation system that displays 3D images in response to changes in the reference markers on fluoroscopic images. This was achieved by specifying the markers on the corresponding fluoroscopic image in the 3D model created preoperatively. This system is intended to track major changes in the object due to surgical procedures. It was successfully in real time tracking of the motion of bone for rotational movement of forearm. Correlations for measurements between the fluoroscopic image and estimated 3D image were excellent. It showed higher accuracy in the anterior-posterior view and posterior-anterior view than in the lateral view. Difficulties were associated with identifying the positional relationship of the markers in the lateral views. This misrecognition was attributed to scale ratio differences in the fluoroscopic images of the front and back or the proximity of the markers in the lateral positions. However, the average differences in the measured values on the fluoroscopic image and 3D bone image for each marker corresponding to the direction of the bone model were 1.1–2.4 mm. This was considered to be close to the accuracy required for navigation (e.g., <2 mm).

There are two potential applications of this technology for clinical practice. It may function as an assistant to decision support in fracture reduction. Surgeons may decide whether reduction is achieved based on comparisons to preoperative planning images. It may also be used to verify implant placement. Surgeons may place an implant in the appropriate position based on preoperative planning images. Due to the challenges associated with recognizing the direction of complex 3D fractures from 2D fluoroscopic images, the surgical procedure under fluoroscopy requires proficiency. Since the successful outcome of fracture reduction surgery has been suggested to depend on accurate and precise intraoperative guidance [29], advances in image-guided surgery, such as 3D intraoperative imaging and surgical tracking, may improve the precision and safety of orthopedic surgery. In order to improve the accuracy of osteosynthesis, it is considered effective to utilize 3D bone morphology and position information such as computer-assisted navigation. To clarify the need for navigation in the osteosynthesis, we conducted a questionnaire survey among local trauma orthopedic surgeons in Japan. As the results, total of 55% surgeons feel the need for navigation (Figure 7). However, various types of fractures, reduction techniques, osteosynthesis materials, and procedures that require rapid treatment have hindered the introduction of navigation in fracture treatment. Currently available navigation for osteosynthesis has problems such as high cost, incompatibility with existing equipment, invasiveness to non-surgical sites, and reduced reliability associated with dynamic motion and morphological changes of surgical manipulations, and is therefore of low practical use. This 3D bone position estimation system is a program that outputs fluoroscopic images to a computer and can be operated on the computer. Therefore, there is no need to introduce a new system. Currently, one of the obstacles to the introduction of navigation systems is the high cost of installation. If this system is put into practical use, it can be expected to reduce the burden on hospitals associated with the introduction of navigation systems. This could facilitate the introduction of preoperative surgical planning into surgery, potentially improving surgical accuracy and safety. The next step should be to evaluate whether this system would improve surgical accuracy.

Several limitations of the tracking algorithm need to be addressed in the future studies. First, since the tracking is based on the last five frames, the tracking of discontinuous images tends to be inaccurate. Second, there is a mismatch for tracking reference points in a straight line in lateral views. This is due to the proximity of the reference markers or the lack of depth information. Third, it may be difficult to place the markers depending on the condition of the surgical site. It is more desirable if alignment can be performed without markers. Fourth, it has also been found to cause discrepancies between fluoroscopy and 3D images when this system is used on large bones such as the pelvis. This is because of the difference in magnifications between the regions near and far from the C-arm and the difference in image distortion between the center and the edge of the radiation field. In addition, to use this system, all of the markers need to be shown under fluoroscopy. Therefore, it is difficult to apply for larger bone. These points should be improved in the future study. These factors result in the misrecognition of positional relationships for the 3D image. We are currently developing a system to adapt discontinuous images and include depth and magnification information.

In conclusion, a 3D bone position estimation system with reference to fluoroscopic images was developed. There were excellent correlations of the reference marker distances between measurements on the fluoroscopic image and estimated 3D image. Marker positions estimated from the fluoroscopic image and the 3D bone image showed higher accuracy in the anterior-posterior view and the posterior-anterior view than in the lateral views. The 3D bone position estimation system can track the rotational motion of the target tissue with an error of less than 3 mm under fluoroscopy. This system helps in real-time comparison of dynamic changes in preoperative 3D and in-traoperative fluoroscopy images.

## Figures and Tables

**Figure 1 diagnostics-12-02237-f001:**
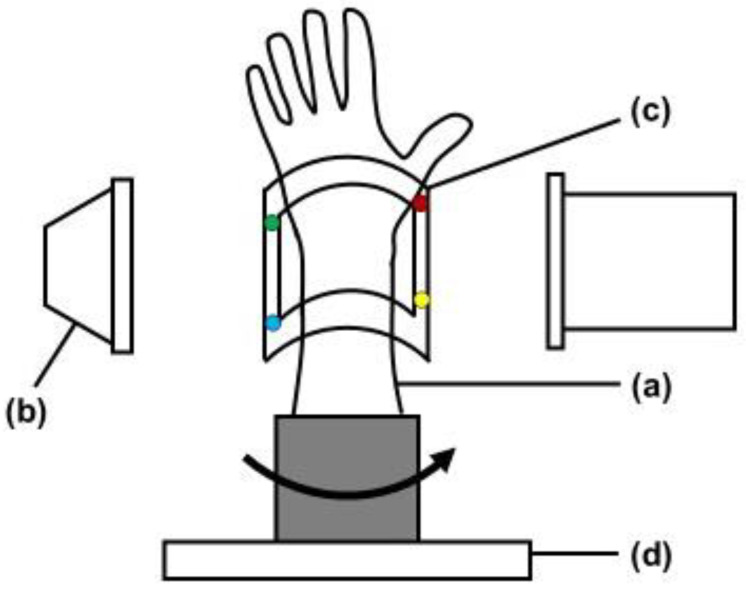
Experimental setting of bone model. (**a**) Bone model. Bone models were covered with X-ray transparent elastic material that imitated skin. (**b**) Position of the image intensifier of fluoroscopy. (**c**) Splint to position the markers. The splint was attached to the bone model. Colored circles indicate the position of each marker. (**d**) Turntable to rotate the bone model.

**Figure 2 diagnostics-12-02237-f002:**
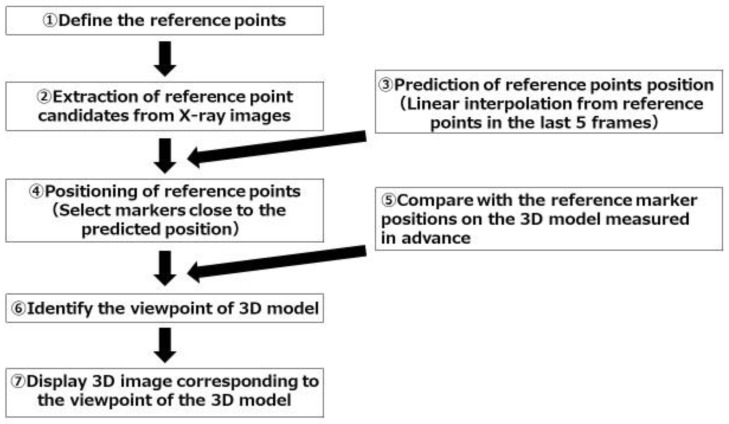
Algorithm of the 3D tracking.

**Figure 3 diagnostics-12-02237-f003:**
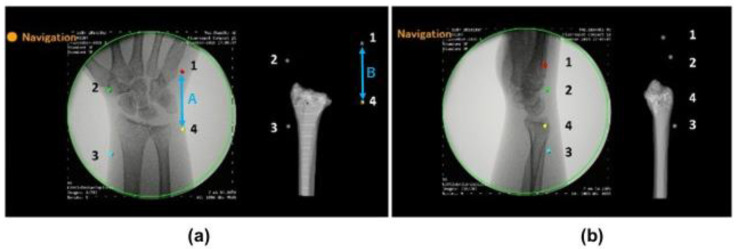
Evaluations of the tracking accuracy. Example images of tracking. (**a**) Posterior-anterior view. (**b**) Ulnar side lateral view. Accuracies were evaluated by comparing the distance between markers on the fluoroscopic image (A) and on the 3D image, which was projected on the monitor (B).

**Figure 4 diagnostics-12-02237-f004:**
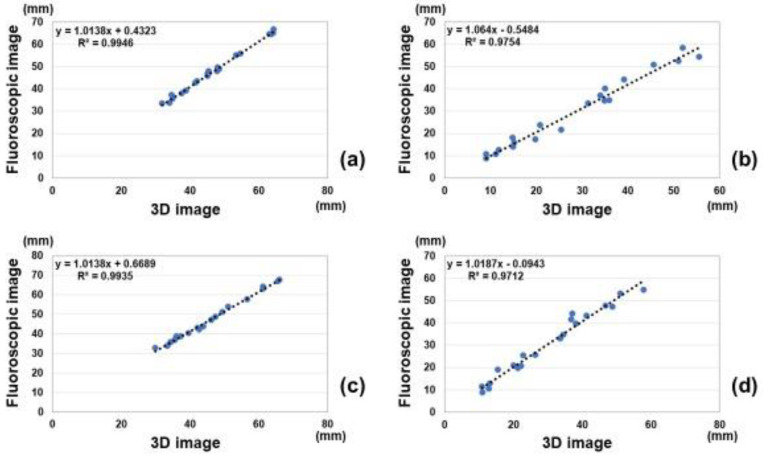
Correlations of marker distances between measurements on the fluoroscopic image and estimated 3D image. (**a**) Anterior-posterior view, (**b**) Ulnar side lateral view, (**c**) Posterior- anterior view, and (**d**) Radial side lateral view.

**Figure 5 diagnostics-12-02237-f005:**
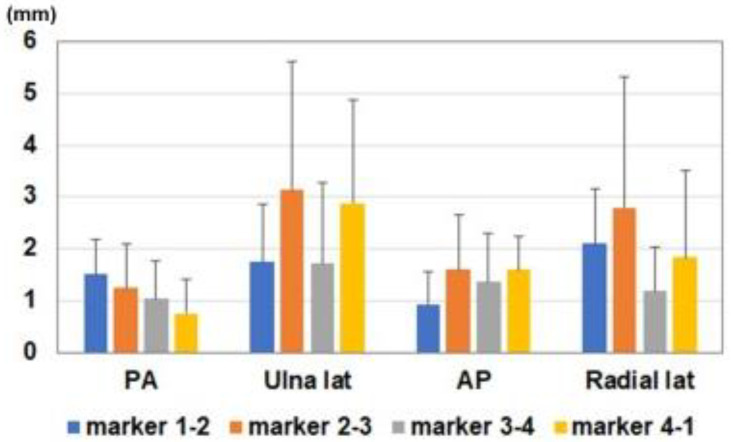
Differences between measured values on the fluoroscopic image and 3D bone image for each marker. Blue bar: difference of the measurement between markers 1-2. Orange bar: difference of the measurement between markers 2-3. Gray bar: difference of the measurement between markers 3-4. Yellow bar: difference of the measurement between markers 4-1.

**Figure 6 diagnostics-12-02237-f006:**
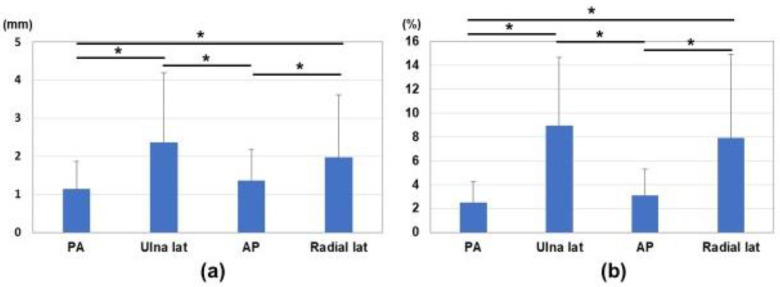
The average differences of the marker distances between measurements on the fluoroscopic image and estimated 3D image. (**a**) Average differences in measured values on the fluoroscopic image and 3D bone image for each marker. (**b**) Ratios of differences in the actual values (measurement difference/actual measurement in the fluoroscopic image). *: *p* < 0.05.

**Figure 7 diagnostics-12-02237-f007:**
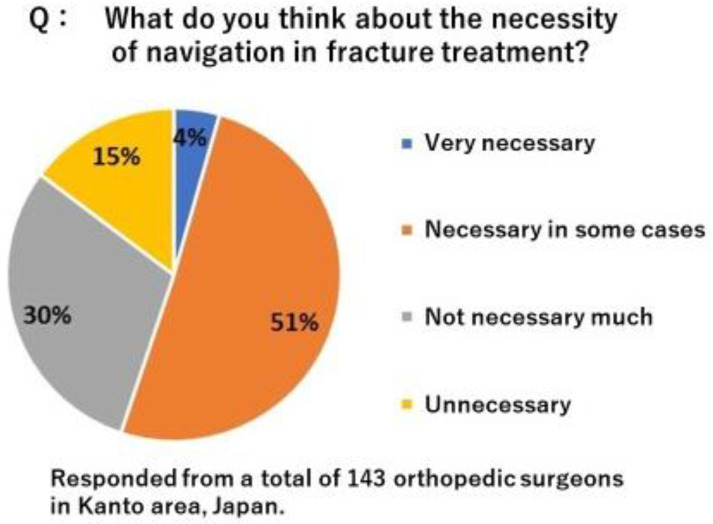
Results of the survey. The necessity of navigation in the fracture treatment.

## Data Availability

The datasets analyzed during the present study are available from the corresponding author upon reasonable request.

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
