# Peer review of "An Experimental Study of a 3D Bone Position Estimation System Based on Fluoroscopic Images"

_diagnostics, 2022, doi:10.3390/diagnostics12092237_

Round 1

Reviewer 1 Report

The authors present an experimental proof of concept for a novel 3D bone positioning system to overcome the difficulties in combining preoperative planning and intraoperative fluoroscopic imaging. The need for such a system is clear and the research design is appropriate. The clinical implications of this study and the direction of further research to translate this method into clinical practice are also discussed in detail. I have two minor suggestions:

1. More details should be provided on the forearm bone models. Did you use sawbones? Also, please indicate how the four conditions were modeled. Did you create the fracture by osteotomy? I assume that the reduction and internal fixation was performed by a trauma surgeon according to clinical practice?

2. I think it should be mentioned more clearly that the study is an experimental setup with a bone model of the radius. After reading the title, I assumed that it was a study in a clinical setting and not limited to a specific site. The transferability of the results to other bones should be discussed.

Author Response

The authors present an experimental proof of concept for a novel 3D bone positioning system to overcome the difficulties in combining preoperative planning and intraoperative fluoroscopic imaging. The need for such a system is clear and the research design is appropriate. The clinical implications of this study and the direction of further research to translate this method into clinical practice are also discussed in detail. I have two minor suggestions:

  1. More details should be provided on the forearm bone models. Did you use sawbones? Also, please indicate how the four conditions were modeled. Did you create the fracture by osteotomy? I assume that the reduction and internal fixation was performed by a trauma surgeon according to clinical practice?

Response)Thank you for the comments. Custom-made bone models were prepared based on CT data of past distal radius fracture cases. With using the CT data, the bone models were made from epoxy resin that could be visualized with fluoroscopy (Kyoto Kagaku Co., Ltd., Japan). A fracture model was created by osteotomy the bone model based on the fracture lines of the previous CT data. An orthopedic surgeon fixed the fracture with plates and wires to create reduction models. These descriptions were added in the text. (Page 2, Line 91-96)

  1. I think it should be mentioned more clearly that the study is an experimental setup with a bone model of the radius. After reading the title, I assumed that it was a study in a clinical setting and not limited to a specific site. The transferability of the results to other bones should be discussed.

Response)Thank you for the comments. Now we clearly stated in the title and text that this is an experimental study. Also, the descriptions for the transferability were added in the text. (Page 1, Line 2, Page 2, Line 89, and Page 9, Line 314-316)

Reviewer 2 Report

Authors proposed a 3D bone position estimation system that displays 3D images in response to changes in the position of fluoroscopic images to compare a 3D preoperative planning image and fluoroscopic image.

The paper is well readable and an accurate results description is provided.

Nevertheless, I think that a description of thr chosen algorithms/techniques to track the poit is missing. In Figure 1 is reported the high level algorithm but no descriprion of the included boxes is provided. The lack of such information impoverish the quality of the paper and prevent the reader to understand how to implement the proposed strategy.

Other comments are listed below:

- L75 if I have well understood, you used a 3D physical phantom to perform the experiment. In this case a description of the used material and design of the 3D structure is missing.

- L96 authors mentioned a program but no details of such program are provided. It reminds me of a text from a patent claim.

- L123 how do you measure the points distance? A common reference system is not described in the manuscript.

- L137 Which projection method did you used?

- L231 authors mentioned a survey questionnaire in a very vague way

- In the Discussion authors report some background, but I'd recommend to move all the state of the art at the beginning of the paper.

- A real incisive conclusion is missing

Author Response

Authors proposed a 3D bone position estimation system that displays 3D images in response to changes in the position of fluoroscopic images to compare a 3D preoperative planning image and fluoroscopic image.

The paper is well readable and an accurate results description is provided.

Nevertheless, I think that a description of thr chosen algorithms/techniques to track the poit is missing. In Figure 1 is reported the high level algorithm but no descriprion of the included boxes is provided. The lack of such information impoverish the quality of the paper and prevent the reader to understand how to implement the proposed strategy.

Response) Thank you for the valuable comments. The tracking algorithm is as follows. 1. CT images of bone models with splint were taken. At the beginning of the experiment, one fluoroscopic image was taken. The program automatically extracts candidate marker points from the image, and associates them with the points on the 3D image as the calibration. Automatic extraction of marker candidate points was performed by image processing. Circles with a certain radius were extracted by Hough transform. 2. Fluoroscopic images were monitored and candidate markers were automatically extracted from each frame. Automatic extractions of candidate markers were performed by the same image processing as in the first step. 3. Circular shapes detected by Hough transform were detected more than the number of metal sphere markers. Therefore, to determine the set of metal sphere markers among them, a linear interpolation of the estimated positions from the last 5 frames was performed. 4. Refer to the result of linear interpolation and determine the nearest circular shape from it as the estimated position of the reference points in the other frame. 5. Compare the positional relationship of the markers on the fluoroscopic image in the current frame with the positional relationship of the reference markers on the 3D model. 6. Calculate the camera pose (orientation, position) that is closest to the positional relationship of the reference markers on the fluoroscopic image. As an index of proximity, the coordinates when a point on the 3D model is projected two-dimensionally with a certain camera pose calculate were calculated, and then the pose that minimizes the squared error of the distance of the reference markers was detected from the fluoroscopic image. 7. Finally, display the 3D image corresponding to the viewpoint of the 3D model. We added these descriptions in the text. (Page 3, Line 130-Page 4, Line 150, and Figure 2)

Other comments are listed below:

- L75 if I have well understood, you used a 3D physical phantom to perform the experiment. In this case a description of the used material and design of the 3D structure is missing.

Response) Thank you for the comments. Custom-made bone models were prepared based on CT data of past distal radius fracture cases. With using the CT data, the bone models were made from epoxy resin that could be visualized with fluoroscopy (Kyoto Kagaku Co., Ltd., Japan). A fracture model was created by osteotomy the bone model, based on the fracture lines of the previous CT data. An orthopedic surgeon fixed the fracture with plates and wires to create reduction models. These descriptions were added in the text. (Page 2, Line 91-96)

- L96 authors mentioned a program but no details of such program are provided. It reminds me of a text from a patent claim.

Response) The program algorithm was described in the text. (Please see the previous response, Page 3, Line 130-Page 4, Line 150, and Figure 2)

- L123 how do you measure the points distance? A common reference system is not described in the manuscript.

Response) The distances between markers were measured using Image J software (NIH, USA). This description was added in the text. (Page 5, Line 170-171)

- L137 Which projection method did you used?

Response) Using image intensifier, the bone model was placed on the turntable and was placed the center of the X-ray output unit. The tracking of the position change of the bone model was verified by rotating the turntable. These descriptions were added in the Figure legend. (Page 3, Line 112-115, Page 3, Line 118-122, and Figure 1)

- L231 authors mentioned a survey questionnaire in a very vague way

Response) Thank you for the comment. We added the descriptions for the survey in the text and showed the results in the Figure 7. (Page 8, Line 283-285, Figure 7)

- In the Discussion authors report some background, but I'd recommend to move all the state of the art at the beginning of the paper.

Response) Thank you for the constructive comment. We moved these descriptions of the background to the introduction section. (Page 2, Line 47-58)

- A real incisive conclusion is missing

Response) We tried to clarify the points of findings in this study and revised the conclusion sentences. We hope that these responses have successfully addressed the reviewer’s point. (Page 8, Line 326-329)